# Specificities in the Structure of the Cartilage of Patients with Advanced Stages of Developmental Dysplasia of the Hip

**DOI:** 10.3390/diagnostics14070779

**Published:** 2024-04-08

**Authors:** Tea Duvančić, Andreja Vukasović Barišić, Ana Čizmić, Mihovil Plečko, Ivan Bohaček, Domagoj Delimar

**Affiliations:** 1Department of Innovative Diagnostics, Srebrnjak Children’s Hospital, 10000 Zagreb, Croatia; tea.duvancic@gmail.com; 2General Hospital “Dr. Anđelko Višić” Bjelovar, 43000 Bjelovar, Croatia; andreja_vukasovic@yahoo.com; 3Sestre Milosrdnice University Hospital Centre, Clinic for Traumatology, 10000 Zagreb, Croatia; anacizmic@gmail.com; 4Department of Orthopaedic Surgery, University Hospital Center Zagreb, 10000 Zagreb, Croatia; mplecko@kbc-zagreb.hr (M.P.); ivan.bohacek@gmail.com (I.B.); 5Department of Orthopaedic Surgery, School of Medicine, University of Zagreb, 10000 Zagreb, Croatia

**Keywords:** developmental dysplasia of the hip, osteoarthritis, neoacetabulum, hyaline cartilage, immunohistochemistry, delayed gadolinium-enhanced MRI of cartilage

## Abstract

Developmental dysplasia of the hip (DDH) presents varying degrees of femoral head dislocation, with severe cases leading to the formation of a new articular surface on the external side of the iliac bone—the neoacetabulum. Despite conventional understanding suggesting otherwise, a tissue resembling hyaline cartilage is found in the neoacetabulum and acetabulum of Crowe III and IV patients, indicating a potential for hyaline cartilage development without mechanical pressure. To test this theory, acetabular and femoral head cartilage obtained from patients with DDH was stained with hematoxylin–eosin and toluidine blue. The immunohistochemical analysis for collagen types II and VI and aggrecan was performed, as well as delayed gadolinium-enhanced MRI of cartilage (dGEMRIC) analysis on a 7.0 T micro-MRI machine. The results obtained from DDH patients were compared to those of the control groups. Hyaline cartilage was found in the neoacetabulum and the acetabulum of patients with DDH. The nature of the tissue was confirmed with both the histological and the MRI analyses. The results of this study proved the presence of hyaline cartilage in patients with DDH at anatomical regions genetically predisposed to be bone tissue and at regions that are not subjected to mechanical stress. This is the first time that the neoacetabular cartilage of patients with advanced stages of DDH has been characterized in detail.

## 1. Introduction

Developmental dysplasia of the hip (DDH) is a disorder of the hip joint, characterized by abnormal development of the hip joint, leading to varying degrees of femoral head dislocation, ranging from mild to complete [1]. DDH is one of the most common causes of secondary hip osteoarthritis (OA). Several radiographical classifications of DDH are described in the literature, such as the Hartofilakidis, Eftekhar, and Crowe classifications [2]. The most commonly used classification is according to Crowe, which classifies DDH in grades I to IV, with grade IV being the most severe. Crowe grade III is characterized by proximal migration of the femoral head by 75–100% and grade IV by proximal migration of the femoral head >100% with respect to the acetabulum [3]. Severe subluxation and luxation of the femoral head result in the formation of a new acetabulum—the so-called neoacetabulum. Patients with Crowe grades III and IV therefore have two distinct acetabular regions—the anatomical acetabulum and the neoacetabulum [2]. It is commonly believed that the normal development of hyaline cartilage requires both genetics and mechanical pressure [4,5,6,7,8]. Patients with DDH grades III and IV, according to Crowe, serve as a unique natural experiment. The cartilage in their anatomical acetabulum develops with the biological potential for development but without mechanical pressure, suggesting the possibility of some type of cartilage tissue development but not hyaline cartilage. Conversely, the neoacetabular region lacks a genetic predisposition for hyaline cartilage formation but experiences significant mechanical pressure. According to the established paradigm, ectopic hyaline cartilage formation should be impossible, and the neoacetabulum should only exhibit bony tissue and potential fibrosis [9,10,11]. Nevertheless, a tissue resembling hyaline cartilage can be found in the neoacetabulum of Crowe III and IV patients. To explore the nature of this tissue, we have advocated that the neoacetabulum might be covered with hyaline cartilage, which migrated there from the anatomical acetabulum as a result of femoral head migration during development [2]. We hypothesized that the structure of the osteochondral unit of the acetabulum in adult patients with DDH corresponds to that of normal cartilage, while the structure of the osteochondral unit of the neoacetabulum and femoral head corresponds to that of the OA.

## 2. Materials and Methods

### 2.1. Patients

Samples were obtained from 3 groups of patients undergoing total hip arthroplasty (THA): (1) patients with DDH-induced secondary OA (DDH group) (*n* = 15; mean age: 53.2; age range: 37–67; 7 f, 8 m); (2) patients with primary OA (OA group, positive control) (*n* = 15; mean age: 58.7; age range: 49–69; 7 f, 8 m); and (3) patients with the femoral neck fracture and otherwise healthy cartilage (Fx group, negative control) (*n* = 15; mean age: 62.5; age range: 53–71; 9 f, 6 m). Exclusion criteria included a history of previously treated fractures, surgeries, or conditions affecting osteochondral unit development, malignancies, autoimmune diseases, and prior corticosteroid treatment. In addition to all the aforementioned conditions, patients from the DDH group had to have been diagnosed with DDH Crowe grades III or IV and OA grades 3 or 4 according to the Kellgren–Lawrence classification, patients from the OA group with OA grades 3 or 4 according to the Kellgren–Lawrence classification, and patients from the Fx group had to show no symptoms or clinical and radiological signs of the OA. This study was approved by the institutions’ Research Ethics Committees. All patients gave informed consent to participate in the study.

### 2.2. Sample Acquisition and Processing

Samples of the osteochondral unit were obtained during the THA procedure using a 10 mm diameter cylindrical chisel (OATS Osteochondral Autograft Transfer System Set, 10 mm, Arthrex, Miami, FL, USA). From the DDH patients, samples were obtained from the anatomical acetabulum (DDH A), the neoacetabulum (DDH NA) [12], the weight-bearing part of the femoral head (DDH WB), and the non-weight-bearing part of the femoral head (DDH NWB). From the control groups, samples were obtained from the acetabulum (OA A and Fx A) and the weight-bearing part of the femoral head (OA F and Fx F). The acetabular samples were obtained from the posterior superior quadrant, according to Wasielewski [13]. The femoral head samples were obtained from zones 3 and 4, according to Ilizaliturri [14]. From each of the aforementioned regions, 2 samples were obtained. One of the samples was fixated in 10% formaldehyde and further processed for histological and immunohistochemical analysis, and the other was stored in a fresh saline solution and scanned on a micro-MRI machine within 48 h following the surgery.

### 2.3. Histology and Immunohistochemistry

After fixation, samples underwent decalcification in 5% formic and 1.5% hydrochloric acid, followed by post-fixation in 4% neutral buffered formaldehyde. Subsequently, histological processing included dehydration in ethanol dilutions, clearing in xylene, and paraffin embedding. Paraffin blocks, cut at 5 μm using a Leitz 1512 Rotary Microtome (Leica Biosystems, Wetzlar, Germany), were mounted on microscope slides.

Slides were then subjected to hematoxylin–eosin and toluidine blue staining after deparaffinization in xylene and rehydration in ethanol dilutions. The progression of the OA was assessed using the Osteoarthritis Research Society International (OARSI) grading system [15].

The immunohistochemical analysis for collagen types II and VI and aggrecan was performed. Slides were deparaffinized in xylene and rehydrated in ethanol. Antigen retrieval for collagen type II was performed using pronase (P8811, Sigma-Aldrich, Taufkirchen, Germany) and hyaluronidase (H3506, Sigma-Aldrich, Taufkirchen, Germany). Antigen retrieval for collagen type VI was performed with overnight incubation in a citrate buffer (pH 6.2). Antigen retrieval for aggrecan was performed with overnight incubation in a citrate buffer (pH 6.2) and subsequent incubation in dithiothreitol (707265ML, Thermo Fisher Scientific, Rockford, IL, USA) and iodoacetamide (A39271, Thermo Fisher Scientific, Rockford, IL, USA). Blocking was performed with 10% goat serum (G9023, Sigma-Aldrich, Taufkirchen, Germany). Samples were incubated with anti-collagen type II (II-II6B3-s, 1:8000, Developmental Studies Hybridoma Bank, Iowa City, IA, USA), anti-collagen type VI (ab199720, 1:1000, Abcam, Cambridge, UK), and anti-aggrecan (12/21/1-C-6-s, 1:150, Developmental Studies Hybridoma Bank, Iowa City, IA, USA) primary antibodies overnight, at 4 °C. Negative control samples were incubated with 1% goat serum. The signal was visualized using the Dako REAL EnVision Detection System (K500711-2, Agilent Technologies, Santa Clara, CA, USA), according to the manufacturer’s instructions. Following covering and drying, the slides were examined under the microscope (CX33, Olympus, Tokyo, Japan) and photographed with the accompanying digital camera (EP50, Olympus, Tokyo, Japan). The level of antigen expression was assessed in terms of saturation levels utilizing ImageJ software (v2.3.0., National Institutes of Health, Bethesda, MD, USA). Saturation was measured on a scale from 0 to 100%, with 0% indicating no detected signal and 100% indicating 100% saturation, i.e., the observed area was completely stained. All measures were normalized to the negative control to eliminate the effect of background staining. The saturation of each sample expressed as % was divided by the saturation of the negative control expressed as %, resulting in a number (no unit) representing the level of expression of each antigen.

### 2.4. Micro-MRI Analysis of Cartilage

Samples were scanned on a 7.0 T micro-MRI machine (BioSpec 70/20 USR, Bruker BioSpin, Ettlingen, Germany) using a standardized protocol (Table 1). Cartilage was scanned using T1 mapping. Samples were then immersed in 0.5 mM gadolinium solution (Dotarem, Pharmacol, Zagreb, Croatia) for 16 h at 4 °C and scanned again using the delayed gadolinium-enhanced MRI of the cartilage (dGEMRIC) protocol.

Micro-MRI scans were analyzed using ImageJ software (v2.3.0.). Cartilage T1 relaxation times before (T1preGd) and after gadolinium application (T1postGd) were calculated. ΔR values of each sample were then calculated as (1/T1postGd-1/T1preGd). Each measurement was taken in triplicates, and mean values were calculated.

### 2.5. Statistical Analysis

Statistical analysis was performed using SPSS Statistics software (v29.0.0.0, International Business Machines Corporation, New York, NY, USA). The data obtained from the same anatomical regions of different experimental groups were compared using the Kruskal–Wallis test, with a significance level of 0.05. If statistically significant differences were found, a post hoc analysis was performed using the Mann–Whitney test.

Statistical analysis of data obtained from different anatomical regions within the DDH group was performed using Friedman’s test and the post hoc Wilcoxon test for dependent samples. Statistical analysis of data obtained from different anatomical regions within the control groups was performed using the Wilcoxon test for dependent samples.

## 3. Results

### 3.1. Microscopic Structure of the Osteochondral Unit

In the DDH group, hyaline cartilage was observed at the neoacetabulum in five cases, as indicated by toluidine blue staining, revealing proteoglycans in the cartilage matrix. This neoacetabular cartilage exhibited structural characteristics resembling hyaline cartilage, including highly organized chondrocytes, minimal hypertrophy, and a columnar organization in the deep zone (Figure 1). The remaining 10 patients in the DDH group did not display hyaline cartilage at the neoacetabulum, though all had fragments resembling damaged hyaline cartilage.

Furthermore, hyaline cartilage was found at the acetabulum in six DDH patients, with toluidine blue staining confirming the presence of proteoglycans (Figure 2). The structure of the acetabular cartilage in these cases resembled hyaline cartilage, while the other nine DDH patients showed no highly organized hyaline cartilage at the acetabulum, yet they exhibited fragments of cartilage tissue.

In the NWB region, in 10 DDH patients, preserved hyaline cartilage was present. This cartilage had the characteristics of healthy hyaline cartilage, with its preserved thickness, highly organized chondrocytes, and no major structural aberrations. The WB region in DDH patients in general showed signs of advanced OA, with highly sclerotic bone either entirely bare or partially covered by degraded cartilage fragments. In four samples, almost entirely healthy cartilage with preserved thickness and organized chondrocytes was observed.

Comparatively, patients with primary OA generally exhibited high stages of OA progression in the osteochondral unit, with the femoral head surface either bare or covered with smaller cartilage fragments. Six patients displayed milder stages of cartilage degradation. In the positive control group, four patients had nearly preserved cartilage on the acetabulum, showing high organizational degrees. The subchondral bone in both the femoral head and acetabulum was predominantly sclerotic.

In the negative control group, the osteochondral unit displayed characteristics of healthy tissue, with both the acetabulum and femoral head covered by healthy hyaline cartilage exhibiting preserved thickness, organized chondrocytes, and minimal surface aberrations. Minimal signs of OA were observed in the subchondral bone, with two patients showing initial stages of cartilage degradation in the acetabulum and femoral head. This degradation manifested as a loss of proteoglycans in the upper layers of the cartilage and damage to the cartilage surface.

### 3.2. Assessment of the Progression of the Osteoarthritis of Cartilage

The level of progression of the OA of cartilage was assessed using the OARSI grading system (Table 2). Statistically significant differences were found between Fx A and OA A (*p* = 0.024), DDH NA and Fx A (*p* < 0.001), Fx F and OA F (*p* = 0.009), DDH WB and Fx F (*p* < 0.001), DDH NWB and DDH WB (*p* = 0.036), DDH NA and DDH NWB (*p* = 0.009, OA A and OA F (*p* = 0.023), and Fx A and Fx F (*p* = 0.005).

### 3.3. Immunohistochemical Analysis of Cartilage

Expression levels of collagen type II, collagen type VI, and aggrecan are shown in Table 3, Figure 3 and Figure 4.

Statistically significant differences between the levels of expression of collagen type II were found between DDH NA and Fx A (*p* = 0.008), and DDH WB and Fx F (*p* = 0.009).

Statistically significant differences between the levels of expression of collagen type VI were found between DDH A and Fx A (*p* = 0.009), DDH WB and Fx F (*p* = 0.005), DDH NWB and Fx F (*p* = 0.003), and Fx A and Fx F (*p* = 0.031).

Statistically significant differences between the levels of expression of aggrecan were found between DDH NA and Fx A (*p* = 0.006), OA A and Fx A (*p* = 0.006), and Fx A and Fx F (*p* < 0.001).

### 3.4. Micro-MRI Analysis of Cartilage

The acetabular and femoral head cartilage T1 relaxation times before and after gadolinium application were measured (Table 4). The relative content of glycosaminoglycans was determined as ΔR values for the acetabular (Figure 5) and the femoral head cartilage (Figure 6). Statistically significant differences in ΔR values were found between Fx A and OA A (*p* = 0.049), DDH NA and Fx A (*p* = 0.001), DDH A and Fx A (*p* < 0.001), DDH NWB and Fx F (*p* = 0.001), DDH WB and Fx F (*p* < 0.001), and Fx A and Fx F (*p* = 0.047). No statistically significant differences were found between any other anatomical regions.

## 4. Discussion

The results of this study proved the presence of hyaline cartilage in patients with developmental dysplasia of the hip in anatomical regions that are genetically predisposed to be bone tissue and in regions that are not subjected to mechanical stress. This is the first time the neoacetabulum of patients with advanced stages of DDH has been characterized in detail.

Hyaline cartilage was found at the neoacetabulum in 33% of DDH patients, as confirmed by positive staining for collagen type II and aggrecan, key hyaline cartilage markers. It expressed collagen type VI primarily in the area of the chondrocyte territory, which is characteristic of healthy hyaline cartilage tissue [6,16,17]. Histological analysis confirmed the structural characteristics of hyaline cartilage. In four subjects with DDH, the neoacetabulum showed positive staining for only collagen type II or aggrecan, suggesting the presence of cartilaginous tissue, likely hyaline cartilage. In the other patients, no cartilage tissue was found on the neoacetabulum. The finding of smaller regions positive for collagen type II or aggrecan in four samples suggests that the neoacetabulum in these patients was covered by cartilaginous tissue, most likely hyaline cartilage, which underwent degeneration due to aging and OA progression. It should be emphasized that all the DDH patients included in the study had been diagnosed with advanced OA, characterized by cartilage degradation. Since preserved hyaline cartilage was found in 33% of the samples, we can conclude that ectopic formation of hyaline cartilage is possible. The existence of hyaline cartilage on the neoacetabulum—an anatomical region that is genetically predisposed to the development of bone tissue—may be attributed to cartilage tissue migration from the anatomical acetabulum during femoral head development. As the femoral head migrates closer to the outer edge of the iliac bone, the cartilage in the natural acetabulum extends in that same direction, creating a neoacetabulum, which is covered with hyaline cartilage [2]. Because of the aforementioned factors, we cannot conclusively claim that hyaline cartilage did not exist in the neoacetabulum of DDH patients where it was not found; rather, we can only speculate as to why it was not found at the time that the research was being conducted: it was destroyed due to degeneration brought on by secondary OA.

Hyaline cartilage was also found on the surface of the acetabulum in 40% of DDH patients. This contradicts the theory requiring mechanical pressure for hyaline articular cartilage development since the anatomical acetabulum’s cartilage is not subjected to such pressure [8]. Despite the lack of articulation of the acetabular cartilage and the femoral head of DDH patients, different stages of tissue degeneration were observed within this group. However, even the samples with advanced stages of tissue degradation had at least fragments of cartilage present, suggesting that the anatomical acetabulum is less susceptible to degenerative changes compared to the neoacetabulum. This claim is also supported by the literature that highlights the role of mechanical wear in tissue degradation [18].

The finding of hyaline cartilage on the neoacetabulum and acetabulum of DDH patients challenges the generally accepted paradigm that both mechanical loading and genetics are necessary for the proper development of hyaline cartilage [4,8,19,20]. Previous studies emphasized the necessity of mechanical pressure in hyaline cartilage development, highlighting its role in regulating extracellular matrix activity, influencing chondrocyte function, and promoting the production of collagen type II and aggrecan [5,21].

Studies report that mesenchymal cells may regulate FGF signaling and the expression of downstream effectors of cartilage differentiation, i.e., sox9 and col2a1 [10]. Commonly, MSCs undergo chondrogenic differentiation in vitro in the presence of TGF-β. [21] It was reported that growth factors (FGF, TGF-β, IGF, BMP-2,-4,-7) are necessary to stimulate chondrogenesis, but they need calcium ions (Ca^2+^) to regulate cell functions, which are provided to cells by channels regulated by mechanical stimuli. [21] Therefore, if adequate mesenchymal cells are located in the neoacetabulum, they might differentiate into cartilaginous tissue [2,10]. On the other hand, our finding of hyaline cartilage on the acetabulum of DDH patients suggests that hyaline cartilage can form even in the absence of mechanical pressure and opens the door to future research.

The osteochondral units of both femoral head regions of DDH patients showed great variability within the group. In 10 DDH NWB samples, completely preserved cartilage with a structure and biochemical composition resembling healthy cartilage was observed. However, in five samples, the cartilage displayed signs of mild-to-severe degeneration, resembling cartilage affected by primary OA in both structure and biochemical composition. This variability in results can be attributed to the fact that DDH-induced secondary OA affects non-weight-bearing regions as well [22,23]. Similar to the acetabulum in DDH patients, the NWB part of the femoral head can be seen as a transitional form from healthy to diseased tissue in the context of this study. While samples from the weight-bearing part of the femoral head in the DDH group mostly exhibited advanced OA, variability was also noted, with four samples showing nearly fully preserved cartilage.

In patients with DDH-induced secondary OA, the cartilage exhibited the highest degrees of degeneration, particularly in weight-bearing regions. dGEMRIC analysis and immunohistochemical staining for collagen type VI revealed significant differences between non-weight-bearing regions of DDH patients and Fx patients. These results agree with existing data indicating varying degrees of degeneration in DDH patients, even in regions not subject to mechanical pressure [23]. Studies on hyaline cartilage have identified the loss of collagen type II as an initial sign of the disease, leading to disorganization of the collagen fiber network and increased water content [16,24,25,26,27,28,29,30]. These changes trigger a cascade of degenerative reactions, including aggrecan loss, abnormal collagen type VI expression and localization, osteophyte formation, subchondral bone sclerosis, and other histopathological alterations. The existence of different degenerative stages in the non-weight-bearing regions of patients with DDH-induced secondary OA suggests a more rapid disease progression compared to primary OA. Additionally, the cascade of degenerative changes may differ between secondary and primary OA. Statistically significant differences in collagen type VI expression and ΔR values between non-weight-bearing regions of DDH patients and negative controls imply that secondary OA caused by DDH initiates with the dysregulation of collagen type VI and glycosaminoglycan expression. Recent emphasis on the role of inflammation in OA progression suggests that damage to non-weight-bearing regions in DDH-induced secondary OA could be explained by inflammatory processes in the affected joint. According to this theory, the expression and diffusion of inflammatory molecules, such as cytokines and chemokines, throughout the hip joint may activate a cascade of degenerative changes, even without the presence of a mechanical stimulus [18,31,32,33,34].

When comparing the cartilage of the acetabular and femoral regions within different groups of patients, it was observed that the femoral regions showed lower degrees of degradation. It is believed that due to the changed morphology, the neoacetabuli of DDH patients have a smaller surface area of the weight-bearing region, resulting in more intense pressure and consequently more severe tissue changes [35]. Similar trends were observed in the control groups, where greater degeneration of the cartilage in the acetabulum compared to the femoral head was observed. These results indicate greater degradation of weight-bearing acetabular regions, even in groups where there are no obvious signs of OA, and are in agreement with the literature published so far.

## 5. Conclusions

This study provided a detailed characterization of the osteochondral unit of the hip joint in patients with advanced stages of DDH. The research confirmed that the weight-bearing regions of DDH patients resemble those affected by primary OA in terms of their structure, while the non-weight-bearing regions of patients with DDH show different degrees of degradation. The most significant finding obtained from this study is the finding of hyaline cartilage on the neoacetabulum and acetabulum of patients with DDH, which once again raises the question of the role of mechanical stimuli and biological potential in the development of hyaline articular cartilage.

## Figures and Tables

**Figure 1 diagnostics-14-00779-f001:**
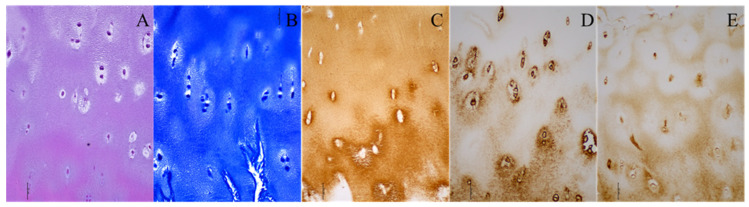
Histological and immunohistochemical staining of the neoacetabular cartilage of a patient with developmental dysplasia of the hip. (**A**) Hematoxylin–eosin; (**B**) toluidine blue; (**C**) collagen type II; (**D**) collagen type VI; (**E**) aggrecan. Images were taken at 200× magnification. The scale bar represents 50 μm. All stainings (**A**–**E**) indicate the presence of hyaline cartilage. Brown (**C**–**E**) indicates the presence of specific hyaline cartilage markers on immunohistochemical stainings.

**Figure 2 diagnostics-14-00779-f002:**
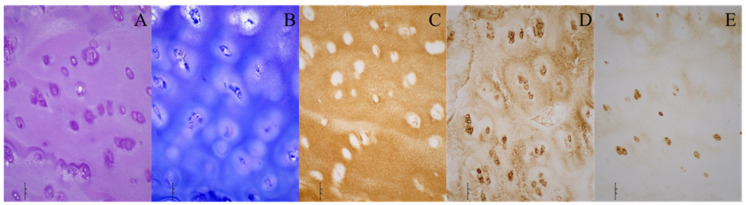
Histological and immunohistochemical stainings of the acetabular cartilage of a patient with developmental dysplasia of the hip. (**A**) Hematoxylin–eosin; (**B**) toluidine blue; (**C**) collagen type II; (**D**) collagen type VI; (**E**) aggrecan. Images were taken at 200× magnification. The scale bar represents 50 μm. All stainings (**A**–**E**) indicate the presence of hyaline cartilage. Brown (**C**–**E**) indicates the presence of specific hyaline cartilage markers on immunohistochemical stainings.

**Figure 3 diagnostics-14-00779-f003:**
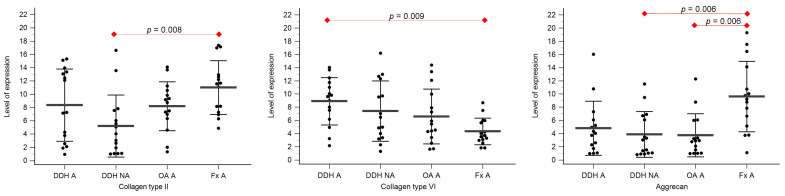
The level of expression of collagen type II, collagen type VI, and aggrecan of the acetabular cartilage of anatomical acetabulum of developmental dysplasia of the hips-induced osteoarthritis group (DDH A); neoacetabulum of developmental dysplasia of the hips-induced osteoarthritis group (DDH NA); acetabulum of the primary osteoarthritis group (OA A); acetabulum of femoral neck fracture group (Fx A). Black dots represent single measurements, while average saturation ± SD is depicted with black lines. Statistical significance is shown on plots with red dots and lines (*p* < 0.05, Mann–Whitney test).

**Figure 4 diagnostics-14-00779-f004:**
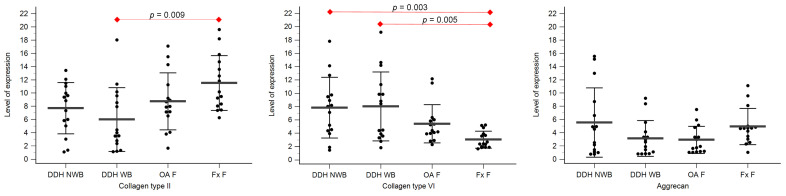
The level of expression for collagen type II, collagen type VI, and aggrecan of the femoral head cartilage of the non-weight-bearing part of the femoral head of developmental dysplasia of the hips-induced osteoarthritis group (DDH NWB), the weight-bearing part of the femoral head of developmental dysplasia of the hips-induced osteoarthritis group (DDH WB), weight-bearing part of the femoral head of the primary osteoarthritis group (OA F), and weight-bearing part of the femoral head of femoral neck fracture group (Fx F). Black dots represent single measurements, while average saturation ± SD is depicted with black lines. Statistical significance is shown on plots with red dots and lines (*p* < 0.05, Mann–Whitney test).

**Figure 5 diagnostics-14-00779-f005:**
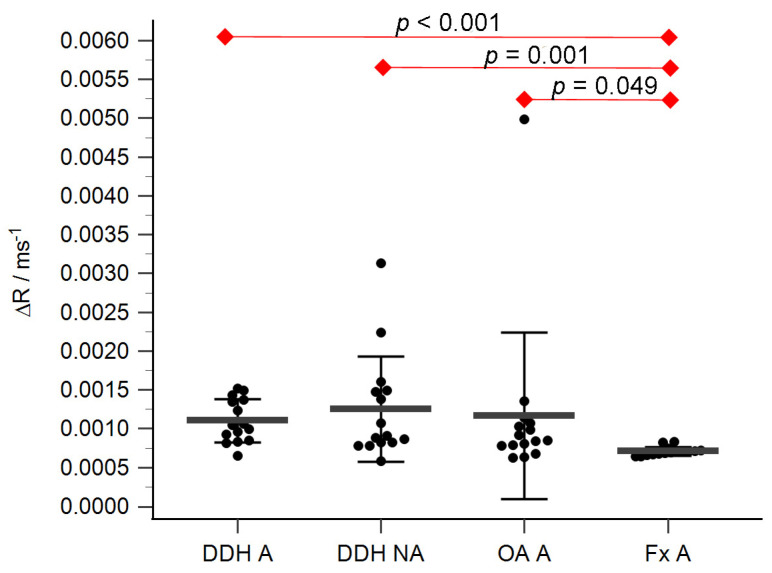
ΔR values of the acetabular cartilage of anatomical acetabulum of developmental dysplasia of the hips-induced osteoarthritis group (DDH A), neoacetabulum of developmental dysplasia of the hips-induced osteoarthritis group (DDH NA), acetabulum of the primary osteoarthritis group (OA A), and acetabulum of femoral neck fracture group (Fx A). Black dots represent single measurements, while average saturation ± SD is depicted with black lines. Statistical significance is shown on plots with red dots and lines (*p* < 0.05, Mann–Whitney test).

**Figure 6 diagnostics-14-00779-f006:**
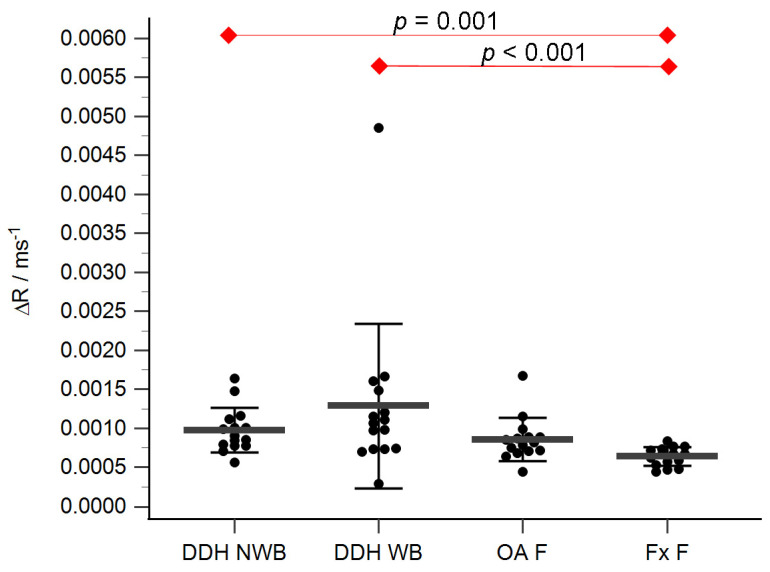
ΔR values of the femoral head cartilage of the non-weight-bearing part of the femoral head of developmental dysplasia of the hips-induced osteoarthritis group (DDH NWB), the weight-bearing part of the femoral head of developmental dysplasia of the hips-induced osteoarthritis group (DDH WB), weight-bearing part of the femoral head of the primary osteoarthritis group (OA F), weight-bearing part of the femoral head of femoral neck fracture group (Fx F). Black dots represent single measurements, while average saturation ± SD is depicted with black lines. Statistical significance is shown on plots with red dots and lines (*p* < 0.05, Mann–Whitney test).

**Table 1 diagnostics-14-00779-t001:** Micro-MRI scanning parameters for T1 maps before and after gadolinium application.

Orientation	Longitudinal/Saggital
FOV	15 × 12 × 5 mm
In-plane resolution	120 × 120 μm
Number of slices	13
Slice thickness/gap	300 μm/100 μm
Sequence type	RARE-VTR sequence with 6 T1 experiments
TR/TE	350, 650, 1000, 1700, 2800, 4000 ms/6 ms
Echo spacing	6 ms
RAREf factor	2
Navg	10
BW	70,000 Hz
Scan time	1 h 28 min

**Table 2 diagnostics-14-00779-t002:** The levels of progression of the osteoarthritis of the acetabular and the femoral head cartilage were assessed by the Osteoarthritis Research Society International (OARSI) grading system. The numbers represent the average OARSI grade ± SD.

	DDH A	DDH NA	OA A	Fx A	DDH NWB	DDH WB	OA F	Fx F
OARSI grade	4.3 ± 2.09	5.2 ± 1.98	4.2 ± 1.56	2.0 ± 0.68	2.6 ± 1.70	5.0 ± 1.70	3.2 ± 1.84	1.2 ± 0.36

Abbreviations: DDH A = anatomical acetabulum of developmental dysplasia of the hips-induced osteoarthritis group; DDH NA = neoacetabulum of developmental dysplasia of the hips-induced osteoarthritis group; OA A = acetabulum of the primary osteoarthritis group; Fx A = acetabulum of femoral neck fracture group; DDH NWB = the non-weight-bearing part of the femoral head of developmental dysplasia of the hips-induced osteoarthritis group; DDH WB = the weight-bearing part of the femoral head of developmental dysplasia of the hips-induced osteoarthritis group; OA F = weight-bearing part of the femoral head of the primary osteoarthritis group; Fx F = weight-bearing part of the femoral head of femoral neck fracture group; OARSI = Osteoarthritis Research Society International.

**Table 3 diagnostics-14-00779-t003:** The level of expression of collagen type II, collagen type VI, and aggrecan for different anatomical regions. The numbers represent average saturation levels ± SD.

Anatomical Region	Collagen Type II	Collagen Type VI	Aggrecan
DDH A	8.36 ± 5.25	8.90 ± 3.46	4.81 ± 3.96
DDH NA	5.21 ± 4.52	7.41 ± 4.41	3.87 ± 3.35
OA A	8.19 ± 3.56	6.59 ± 4.01	3.73 ± 3.15
Fx A	11.03 ± 3.91	4.33 ± 1.95	9.61 ± 5.16
DDH NWB	7.71 ± 3.74	7.84 ± 4.41	5.54 ± 5.06
DDH WB	6.00 ± 4.66	8.04 ± 5.00	3.15 ± 2.61
OA F	8.75 ± 4.16	5.42 ± 2.78	2.92 ± 1.99
Fx F	11.51 ± 3.99	3.04 ± 1.23	4.94 ± 2.64

Abbreviations: DDH A = anatomical acetabulum of developmental dysplasia of the hips-induced osteoarthritis group; DDH NA = neoacetabulum of developmental dysplasia of the hips-induced osteoarthritis group; OA A = acetabulum of the primary osteoarthritis group; Fx A = acetabulum of femoral neck fracture group; DDH NWB = the non-weight-bearing part of the femoral head of developmental dysplasia of the hips-induced osteoarthritis group; DDH WB = the weight-bearing part of the femoral head of developmental dysplasia of the hips-induced osteoarthritis group; OA F = weight-bearing part of the femoral head of the primary osteoarthritis group; Fx F = weight-bearing part of the femoral head of femoral neck fracture group.

**Table 4 diagnostics-14-00779-t004:** T1 relaxation times before (T1preGd) and after gadolinium application (T1postGd), and ΔR values of different anatomical regions.

Anatomical Region	T1preGd/ms	T1postGd/ms	ΔR/ms^−1^
DDH A	1823.87 ± 204.12	615.64 ± 89.16	1.10 × 10^−3^ ± 2.67 × 10^−4^
DDH NA	1706.73 ± 322.61	583.04 ± 140.60	1.26 × 10^−3^ ± 6.56 × 10^−4^
OA A	1634.37 ± 157.78	641.65 ± 142.76	1.17 × 10^−3^ ± 1.04 × 10^−3^
Fx A	1464.61 ± 72.55	718.35 ± 25.69	7.01 × 10^−4^ ± 5.41 × 10^−5^
DDH NWB	1651.39 ± 217.34	643.33 ± 96.87	9.76 × 10^−4^ ± 2.75 × 10^−4^
DDH WB	1625.76 ± 265.37	596.01 ± 162.16	1.29 × 10^−3^ ± 1.02 × 10^−3^
OA F	1625.85 ± 260.06	686.59 ± 91.36	8.58 × 10^−4^ ± 2.68 × 10^−4^
Fx F	1432.54 ± 79.19	754.88 ± 72.95	6.37 × 10^−4^ ± 1.16 × 10^−4^

Abbreviations: DDH A = anatomical acetabulum of developmental dysplasia of the hips-induced osteoarthritis group; DDH NA = neoacetabulum of developmental dysplasia of the hips-induced osteoarthritis group; OA A = acetabulum of the primary osteoarthritis group; Fx A = acetabulum of femoral neck fracture group; DDH NWB = the non-weight-bearing part of the femoral head of developmental dysplasia of the hips-induced osteoarthritis group; DDH WB = the weight-bearing part of the femoral head of developmental dysplasia of the hips-induced osteoarthritis group; OA F = weight-bearing part of the femoral head of the primary osteoarthritis group; Fx F = weight-bearing part of the femoral head of femoral neck fracture group.

## Data Availability

The datasets used and/or analyzed during the current study are available from the corresponding author on reasonable request.

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
