# Peer review of "Specificities in the Structure of the Cartilage of Patients with Advanced Stages of Developmental Dysplasia of the Hip"

_diagnostics, 2024, doi:10.3390/diagnostics14070779_

Round 1

Reviewer 1 Report

Comments and Suggestions for Authors

The authors have carried out an interesting study that suits the readership of the journal. However the following points will improve the readability of the manuscript.

1. Please expand the acronyms as footnotes in all Tables.

2. Represent the mean, SD or SE values in all Figures.

3. Provide legends for all figures indicating the key messages including statistically significant differences.

Reviewer 2 Report

Comments and Suggestions for Authors

- I recommend using a more scientific approach towards your graphs and representations of results; the visual representations you've chosen is the basic one provided by Microsoft Excel and is at a student level;

- the manuscript lacks a conclusion section;

- authors should acknowledge the existence of other classification systems and justify the focus on the Crowe classification only;

- include some of the factors that are influencing cartilage development, such as biochemical factors and others;

- what were the assumptions underlying each statistical test employed?

- justify the choice of micro-MRI and dGEMRIC protocol over other techniques;

- what were the scales used for quantifying antigen expression levels from immunohistochemistry;

- references are up to date;

- the title is adequate;

Round 2

Reviewer 2 Report

Comments and Suggestions for Authors

Paper is now ready and looks more scientific. Authors have made the changes required.